# Controlling the Architecture of Freeze-Dried Collagen Scaffolds with Ultrasound-Induced Nucleation

**DOI:** 10.3390/polym16020213

**Published:** 2024-01-11

**Authors:** Xinyuan Song, Matthew A. Philpott, Serena M. Best, Ruth E. Cameron

**Affiliations:** Department of Materials Science and Metallurgy, University of Cambridge, Cambridge CB3 0FS, UK; xs284@cam.ac.uk (X.S.); smb51@cam.ac.uk (S.M.B.)

**Keywords:** freeze-drying, collagen scaffolds, ice nucleation, ultrasound, porous architecture, tissue engineering

## Abstract

Collagen is a naturally occurring polymer that can be freeze-dried to create 3D porous scaffold architectures for potential application in tissue engineering. The process comprises the freezing of water in an aqueous slurry followed by sublimation of the ice via a pre-determined temperature–pressure regime and these parameters determine the arrangement, shape and size of the ice crystals. However, ice nucleation is a stochastic process, and this has significant and inherent limitations on the ability to control scaffold structures both within and between the fabrication batches. In this paper, we demonstrate that it is possible to overcome the disadvantages of the stochastic process via the use of low-frequency ultrasound (40 kHz) to trigger nucleation, on-demand, in type I insoluble bovine collagen slurries. The application of ultrasound was found to define the nucleation temperature of collagen slurries, precisely tailoring the pore architecture and providing important new structural and mechanistic insights. The parameter space includes reduction in average pore size and narrowing of pore size distributions while maintaining the percolation diameter. A set of core principles are identified that highlight the huge potential of ultrasound to finely tune the scaffold architecture and revolutionise the reproducibility of the scaffold fabrication protocol.

## 1. Introduction

Collagen is a natural polymer that can be processed to form three-dimensional porous scaffolds. These structures are used widely as substrates for tissue engineering, providing mechanical support for cell attachment and stimulating desirable cellular interactions [1]. Freeze-drying, also known as lyophilisation, is a popular process for producing these types of scaffolds. An aqueous slurry of collagen is frozen, creating an interlinked network of ice crystals with the collagen excluded to the edges of the crystals [2]. Then, the pressure is reduced, and the temperature is raised to induce sublimation, creating a porous scaffold with pores replacing the ice crystals. Therefore, the initial freezing process should be controlled carefully to form desirable ice crystal morphology and final scaffold architecture [3,4].

The solidification of ice is a two-stage process which involves ice nucleation and crystal growth. Nucleation proceeds with the initial formation of a crystal when the temperature is lower than the equilibrium freezing temperature (T_E_). Below T_E_, the system appears in a metastable state where many clusters of different radii may form, but without a long-range order [5,6]. Under the Classical Nucleation Theory, each nucleus encounters an energetic barrier to formation due to factors including loss of entropy, increased strain energy and increased interfacial energy at the solid–liquid interface [7,8]. Only crystals larger than the critical radius may form a nucleus and grow spontaneously. Primary nucleation can be either homogeneous or heterogeneous. Heterogenous nucleation takes place in the presence of solutes, impurities, grain boundaries or dislocations, reducing the energy barrier for nucleation [9]. Afterwards, secondary nucleation can proceed when new crystals are produced from pre-existing crystals [10].

The temperature and location at which nucleation occurs have significant impact on the architecture of the ice-templated scaffold. Scaffolds of 0.5 wt% collagen have significantly larger pore sizes than those produced from 1 wt% collagen [11]. It is believed that higher collagen content provides additional surface area and acts as a nucleation site for heterogeneous nucleation, forming many small ice crystals and replicating scaffolds with smaller pores [4,12,13]. The thermal gradient within the sample at nucleation influences the isotropy of scaffold architecture. If there is a large thermal gradient inside the slurry and regions of the slurry stay above the equilibrium freezing temperature at nucleation, an anisotropic structure can be produced [4,11]. In practical terms, uneven slurry cooling can be achieved by increasing the filling height of the slurry and hence distance from the cooling shelf or changing the mould material to have a significantly different thermal conductivity from the slurry [4,11].

After nucleation, the crystal growth proceeds, and the growth rate can be affected by the effectiveness of latent heat removal [14,15]. Pawelec et al. introduced a thermal parameter, “time at equilibrium”, defined as the time that the slurry spends around the equilibrium freezing temperature [16]. The cut-off point for the equilibrium temperature was set as −1.5 °C when large-scale molecular movement during the ice–water transition process is completed. The pore size of an ice-templated collagen scaffold is closely correlated with the time at equilibrium [16]. There are a number of practical ways to affect the efficiency of latent heat release in a freezing slurry and therefore lengthen the time at equilibrium: a higher cooling shelf temperature is set, a lower collagen concentration is used and the mould contact area with the heat sink is reduced [4,11]. Moreover, with ineffective latent heat removal, crystallisation proceeds slowly, allowing annealing before the end of solidification [17]. In annealing, also known as Ostwald ripening, small crystals are dissolved and incorporated into larger ones, which leads to the formation of larger crystals.

Tuning the processing parameters during freeze-drying can tailor the scaffold architecture for different applications. However, scaffolds fabricated in the same batch or under identical external thermal conditions may still exhibit differing architectures as the result of the stochastic nature of ice nucleation [4,18,19,20]. Figure 1 summarises the key arguments explaining how the stochastic ice nucleation leads to large scaffold structural variations. Controlling the temperature of nucleation would allow greater control of scaffold reproducibility. The addition of nucleation agents can narrow the range of temperature at which nucleation occurs [19]. Also, multiple methods to stimulate nucleation have been explored, including seeding, mechanical methods (shaking, taping and agitation), and the applications of ultrasound, laser, electric and magnetic fields [21,22,23,24,25]. Ultrasound has emerged as particularly promising due to its low cost, ease of operation, and avoidance of ionising radiation, high temperature and direct contact with the sample.

Ultrasound is defined as an acoustic wave with a frequency greater than 20 kHz [26]. When ultrasound is applied to a liquid, zones of low and high pressure are generated, and microscopic voids are formed in the liquid in a process known as cavitation [26,27,28]. Bubbles that experience periodic growth and compression over many cycles are defined as stable (or repetitive) cavitation bubbles [29,30]. In contrast, bubbles which grow and collapse violently within a small number of cycles are known as unstable (transient) cavitation bubbles [31].

Ultrasound vibration has been shown to facilitate the phase transition from supercooled water to ice [32]. According to Hickling’s theory, vigorous collapsing of cavitation bubbles can create local zones of high pressure exceeding 5 GPa over a period of nanoseconds resulting in high degrees of supercooling, which acts as the driving force for nucleation [27]. Furthermore, ultrasound can break down pre-existing ice dendrites into smaller fragments, initiating secondary nucleation [32,33]. Zhang et al. suggested the flow streams caused by the cavitation bubbles may promote nucleation [34]. Dodds et al. suggested that molecular aggregates form due to the pressure gradient around microbubbles, causing nucleation [35]. The exact mechanism is still under debate, and it is likely that multiple mechanisms may be involved in the process of ultrasound-induced nucleation.

Ultrasound technologies have been applied in the fields of biomaterial fabrication and drug development, improving precision and reproducibility [30,36,37,38]. In 2022, Xiao et al. reported the use of an ultrasound-assisted freeze-drying technique to fabricate polyimide aerogels, where the ultrasound was applied to the gel during the entire freezing phase. Aerogel prepared using this new technique had smaller pores and a more consistent structure compared with that prepared using the traditional method. The pore structure can be further controlled by tuning ultrasonic power [39]. Kiani et al. explored the effect of ultrasound on ice crystal formation in frozen agar gels. Their finding suggests that the temperature to apply ultrasound to the supercooled slurry was the key to tailoring the ice crystal size, while no major difference was observed when tuning the ultrasound power, exposure time and duty cycle [40].

In this study, ultrasound was incorporated into the standard freeze-drying protocol to fabricate collagen scaffolds for tissue engineering applications. The hypothesis leading to this work was that using ultrasound to trigger nucleation at a particular temperature can improve the control over scaffold architecture and the reproducibility of the scaffold fabrication process. This work aimed to first explore the stochastic nature of ice nucleation, measuring the nucleation temperature of collagen slurries of 0.5, 1 and 1.5 wt%. Then, the ability of ultrasound to trigger ice nucleation in 1 wt% collagen slurry across a range of temperatures was explored. Finally, the impact of the ultrasound-induced nucleation method on the thermal profiles of the freezing slurry and resultant scaffold architecture was investigated. The knowledge gained from this study can serve as a fundamental guideline for fabricating ice-templated scaffolds with ultrasound to form desirable structures.

## 2. Materials and Methods

### 2.1. Collagen Slurry Preparation and Thermocouples Set-Up

Collagen suspensions of 0.5, 1 and 1.5 wt% were prepared by hydrating collagen powder, type I collagen derived from Dervo bovine dermal collagen (Collagen Solutions, Glasgow, UK), in 0.05 M acetic acid (Sigma-Aldrich, Dorset, UK). The suspension was left in a 4 °C fridge for at least 72 h for hydration. Then, the suspension was blended at 18,000 rpm for 2 min, left to rest for 1 min and blended again at 22,000 rpm for 1 min. Afterwards, the slurry was centrifuged at 2500 rpm for 5 min to remove air bubbles.

CytoOne 12-well cell culture plates were used as moulds. For each well, a type K thermocouple (Pico Technology, Saint Neots, UK) was fixed at 10 mm height of the well wall. Thermocouples were then plugged into a portable temperature data logger (Omega, Manchester, UK). Next, 4 mL of collagen slurry was pipetted into each well of the plate and left in the fridge until the temperature stabilised.

### 2.2. Random Nucleation of Ice

The stochastic nucleation temperatures of collagen slurries with different concentrations were measured. The cell culture plate containing collagen slurry was placed into a −20 °C spark-free laboratory-grade chest freezer (WolfLabs, York, UK), and the slurry was left to freeze naturally. The temperature profile was measured using a type K thermocouple and recorded by a temperature data logger. The nucleation of ice was identified by a rapid increase in temperature to 0 °C on the release of latent heat. Nucleation temperature was defined as the last temperature recorded before a characteristic jump in temperature reading, as shown in Figure 2.

### 2.3. Scaffold Fabrication

In this study, 1 wt% collagen slurry was used for scaffold fabrication. As shown in Figure 3, collagen scaffolds were produced using a two-step freeze-drying process: freezing in a precooled commercial ultrasonic bath inside a −20 °C freezer, followed by drying in a VirTis AdVantage freeze dryer (Biopharma Process Systems, Winchester, UK). During the freezing process, the temperature profile of the slurry was recorded, identifying the nucleation temperature and time at equilibrium, as illustrated in Figure 2. The time at equilibrium, defined by Pawelec et al., measures the time the slurry spent above the equilibrium cut-off temperature (−1.5 °C) [16].

Two different ice nucleation methods were applied in the initial freezing step. (1) Ultrasound-induced nucleation: An ultrasonic cleaning bath (40 kHz frequency and 0.2 Wcm^−2^ intensity) was placed inside the −20 °C freezer. To reduce the attenuation effect of ultrasound, the bath was filled with 200 mL ethanol (50% *v*/*v*). Once the temperature inside the ultrasonic bath stabilised to around −16 °C, the cell culture plate with collagen slurry was transferred from the 4 °C fridge into the ultrasonic bath. The temperature at the top of the slurry was carefully monitored. Ultrasound was switched on for 3 s to induce ice nucleation at the chosen test temperature while minimising potential thermal effects. Samples were then left to freeze for more than 60 min. (2) Random nucleation: A similar experimental setup was used, but no ultrasound was applied, and the slurry was left to freeze naturally, with the temperature profile recorded.

The freeze dryer shelf was manually cooled to −20 °C, and the frozen collagen slurry was transferred to a freeze dryer for the drying step. The pressure was reduced to 80 mTorr over 30 min, the temperature increased to 0 °C over 30 min, and these conditions were held for 1200 min for ice sublimation. The scaffold fabrication process was repeated for both ultrasound-induced and random nucleation groups to obtain samples nucleating across a wide range of temperatures.

### 2.4. Scanning Electron Microscopy (SEM)

SEM was used to visualise the morphology of the scaffold in 2D. Horizontal and vertical cross-sections were cut from the centre of the scaffold using a scalpel. Then, samples were mounted on the sample holder using carbon tape and sputter-coated with a layer of platinum for 2 min at 40 mA. Scaffolds were imaged using the FEI Nova Nano SEM system (FEI Company, Hillsboro, OR, USA) under a secondary electron mode, with an aperture size of 30 μm, an operating voltage of 5 kV and a spot size of 3 nm.

### 2.5. Micro-Computed Tomography (Micro-CT)

Three-dimensional quantitative structural characterisation of scaffolds was carried out using micro-CT. Offeddu et al. suggested little difference in the pore architecture between dry and hydrated scaffolds, yet higher background noise was observed during the scanning of hydrated scaffolds [41]. Therefore, in this study, dry scaffolds were directly characterised without hydration. Cylindrical samples were cut from scaffolds using an 8 mm biopsy punch. Then, they were imaged under a micro-CT (Skyscan 1272, Bruker, Kontich, Belgium) with an operating voltage of 25 kV, a current of 93 μA and a voxel size of 3 μm^3^ (5 μm^3^ for scaffolds with large pores). The scan was performed under 180° rotation, with a rotation step size of 0.2 and an exposure time of 1830 ms. Three-dimensional image reconstruction was carried out using NRecon (Version 1.6.9.8, Bruker, Kontich, Belgium) with a threshold range from 0 to 0.06. The scaffold architecture was visualised in 3D using the program DataViewer (Version 1.5.0, Bruker, Kontich, Belgium). Afterwards, the dataset of each scaffold was analysed using CTAnalyser (Version 1.20.3.0, Bruker, Kontich, Belgium). Cubic volumes of interest (VOIs) of 1 mm^3^ were selected from each sample. The images were thresholded under the Otsu method and de-speckled using the sweep function. Three-dimensional structural analysis was performed to measure the scaffold pore size and degree of anisotropy. Pore size was calculated using a sphere fitting algorithm, which is defined as the largest diameter of a sphere which can be bounded by a pore [42]. The degree of anisotropy is defined as the ratio between the maximum and the minimum radius of the mean intercept length (MIL) [43]. MIL distribution was calculated by overlaying parallel lines in different directions on the 3D reconstructed data.

The percolation diameter is a scale-invariant property describing the transport pathway through the scaffold [44]. It is defined as the size of the largest sphere able to travel through an infinitely large scaffold without obstructions. The shrink wrap feature in CTAn was used to identify the volume accessible to a virtual sphere of different sizes. From Equation (1), for objects with different diameters (d) invading the scaffold, their linear penetration depths in the z-direction (L) can be measured. Then, by extrapolating the line to an infinite sample size, the percolation diameter (d_c_) was obtained. In this study, the bitmaps saved from shrink wrap operation were directly analysed using the “segmented percolation method” developed by Nair et al. which excludes the anomalous features within the dataset and calculates percolation diameter using a bespoke Python script [45].
(1)$$\;L=L_{0}{\left(d-d_{c}\right)}^{-0.88}.$$

## 3. Results

### 3.1. Nucleation Temperature Distribution in Collagen Slurry

The random nucleation temperatures of 0.5, 1.0 and 1.5 wt% collagen slurries were measured and organised into histograms, as shown in Figure 4. The nucleation of ice in collagen slurries was found to occur across a broad range of temperatures.

### 3.2. Thermal Profile of Freezing Collagen Slurry

For the following study, 1 wt% collagen slurry was chosen due to its wide applicability in scaffold fabrication. Ultrasound was applied to the supercooled collagen slurry to stimulate ice nucleation at different temperatures. Figure 5 shows the thermal profiles of four representative samples with ultrasound-triggered ice nucleation at −1.7, −3.8, −5.8 and −10.1 °C, respectively. The onset of nucleation was characterised by an instantaneous jump in temperature to around the equilibrium freezing temperature.

### 3.3. Scaffold Architecture Visualisation

Scaffolds templated by ice that nucleated at different temperatures were visualised using micro-CT and SEM in a dry state. As shown in Figure 6, with a more negative nucleation temperature, the scaffolds displayed smaller, more isotropic pores with a narrow distribution of sizes. If the ice nucleated at a temperature above about −4 °C, a coarse, anisotropic, plate-like structure was observed. Scaffolds with ultrasound-induced nucleation had smaller pores than those nucleated randomly. A smooth pore wall was observed in scaffolds produced under both nucleation methods.

### 3.4. Relating Scaffold Architecture to Thermal Parameters

Figure 7A shows the time at equilibrium plotted against nucleation temperature. For random nucleation (RN), the slurry tended to spend less time at equilibrium if it nucleated at a more negative nucleation temperature. In comparison, for ultrasound-induced nucleation (UIN), the time at equilibrium stayed broadly constant with different nucleation temperatures. For nucleation temperature above about −6 °C, collagen slurries with RN spent a significantly longer time at equilibrium than those with UIN, as illustrated in Figure 7B,C.

The average scaffold pore size was plotted against nucleation temperature and time at equilibrium for the two nucleation methods. As shown in Figure 8A, there was clearly a positive correlation between the average pore size of the scaffold and nucleation temperature. With UIN, the scaffolds generally had a smaller pore size across the entire nucleation temperature range. The difference in scaffold pore size between the two nucleation methods was found to be more significant with small undercooling. The average scaffold pore size was also plotted against time at equilibrium in Figure 8B. For the RN group, scaffolds generally had a larger pore size if the slurry spent longer time at equilibrium during freezing. For the UIN group, the data points centred around smaller time at equilibrium and pore size.

The pore size distribution of collagen scaffolds produced with ice nucleating at different temperatures was plotted in Figure 9. For samples with a lower ice nucleation temperature, the width of the pore size distribution was smaller. With UIN, the scaffolds had a narrower distribution of pore sizes, implying the formation of a more uniform, consistent pore architecture.

Figure 10 depicts the average scaffold degree of anisotropy with respect to ice nucleation temperature. Scaffolds usually had a smaller degree of anisotropy if ice nucleated at a lower temperature, implying the formation of a more isotropic architecture. No significant difference was observed between RN and UIN groups.

The results of scaffold percolation diameter versus nucleation temperature are shown in Figure 11. The general trend suggests that at a lower nucleation temperature, scaffolds tended to have smaller percolation diameters. However, there was a relatively large amount of scatter in the data and no significant difference was observed between RN and UIN groups.

## 4. Discussion

### 4.1. Stochastic Nature of Ice Nucleation

Figure 4 reveals that the nucleation temperatures in collagen slurries of different concentrations were spread over a wide range of values, which confirmed the large variability in ice nucleation temperature under identical conditions. The random nucleation event then led to large variations in scaffold architecture, as depicted in Figure 6. The distribution of nucleation temperatures in 1 wt% collagen slurry was centred around a higher temperature than samples based on 0.5 wt% slurry. The higher concentration of solids provided more nucleation sites, reducing the amount of undercooling required for nucleation. However, in 1.5 wt% collagen slurries, the increased viscosity was likely to have suppressed the formation of nuclei [46]. This effect might have outweighed the effects of the presence of increased nucleation sites, shifting the nucleation events to higher degrees of undercooling. This finding highlights the need to control the ice nucleation event for reproducibility in scaffold fabrication.

### 4.2. Ultrasound-Triggered Nucleation

The application of ultrasound to stimulate ice nucleation in a supercooled 1% collagen slurry is illustrated in Figure 5, resulting in an instantaneous increase in slurry temperature to the equilibrium freezing temperature. Therefore, the effectiveness of ultrasound-induced nucleation in collagen slurry was confirmed. This could be explained by Hickling’s theory, as depicted in Figure 12: the application of ultrasound leads to the formation of microbubbles [27]. Then, the fluctuating ultrasound wave causes the collapse of cavitating bubbles. This generates local zones of high pressure which raises the equilibrium freezing temperature of the water, providing the driving force of ice nucleation. Also, the flow streams formed due to the movement of stable cavitation bubbles can promote nucleation [34]. This finding was later confirmed by Chow et al. with microscopy studies [33]. In addition, Dodds et al. believed that the huge pressure gradient generated by an oscillating bubble helps the diffusion of particles, forming large molecular aggregates. Thus, the cavitation bubbles acted as a cluster attachment reactor, facilitating the formation of a nucleus [35,47]. It is likely that multiple mechanisms play a role in the process of ultrasound-induced nucleation.

### 4.3. Ultrasound-Induced Nucleation and Random Nucleation: Correlating Thermal Parameters to Scaffold Architecture

#### 4.3.1. Pore Morphology

For nucleation temperatures ranging from 0 to −4 °C, large anisotropic plate-like structures were formed. At lower nucleation temperatures, scaffolds tended to have smaller and equiaxed pores. There was no significant difference in pore morphology between the two nucleation methods. These findings were consistent with those of Nakagawa et al. reporting that in 10% frozen mannitol solutions, large, dendritic ice crystals were found in samples nucleated at higher temperatures, whereas smaller and more uniform ice crystals were presented in samples nucleated at lower temperatures. They believed that the degree of undercooling affects the number of nuclei formed in the slurry, determining final crystal morphology [20].

#### 4.3.2. Time at Equilibrium against Nucleation Temperature

Time at equilibrium and nucleation temperature are two key thermal parameters during the slurry freezing process. The nucleation temperature measures the amount of undercooling before crystal nucleation. Time at equilibrium reflects how long the slurry spends in the crystal growth phase [16]. As shown in Figure 7, for the random nucleation group, the slurry which nucleated at a lower nucleation temperature tended to have a shorter time at equilibrium. With a larger undercooling, there is a larger driving force for nucleation. Therefore, as more nuclei are present, the crystal growth phase is shortened due to the limited space available [16]. However, with ultrasound, the time at equilibrium remained largely constant across a wide range of nucleation temperatures. With a small undercooling, the ultrasound effectively promotes the formation of nuclei, reducing the time required for subsequent crystal growth. With a larger undercooling, the number of nuclei did not increase further with ultrasound. As a result, the time at equilibrium did not reduce further upon the application of ultrasound.

#### 4.3.3. Pore Size against Nucleation Temperature and Time at Equilibrium

As shown in Figure 8A, scaffolds had a smaller average pore size if ice nucleated at a lower temperature. At a greater undercooling, there is a larger driving force for nucleation and a smaller driving force for growth [48]. Therefore, the ice crystals are smaller, replicating a scaffold with smaller pores. Compared with random nucleation, slurries that underwent ultrasound-induced nucleation generally exhibited smaller average pore sizes. The difference between the two nucleation methods was more obvious with a smaller undercooling. The application of ultrasound could introduce more ice nuclei simultaneously in the supercooled sample, which limited the space for crystal growth [26,49]. Therefore, many small ice crystals formed, leading to scaffolds with smaller pore sizes.

As shown in Figure 8B, for random nucleation, the scaffolds had smaller pore sizes if the slurry spent a shorter time at equilibrium. This was in good agreement with the results of Pawelec et al., who suggested a universal relationship between the scaffold pore size and time at equilibrium [16]. With ultrasound, more nucleation sites were formed, shortening the time required for subsequent crystal growth [50]. Therefore, a shorter time at equilibrium and smaller average pore sizes were observed for samples with ultrasound-induced nucleation.

#### 4.3.4. Pore Size Distribution

Scaffolds produced by freeze-drying naturally contain a distribution of pores with different sizes. In Figure 9, it can be seen that as nucleation temperature reduces, the scaffold tends to have a smaller distribution or pore sizes, suggesting a more homogenous structure. Scaffolds formed with random and ultrasound-induced nucleation of ice at a similar nucleation temperature range were compared. The pore size distribution was centred around a smaller value and was narrower for samples formed with ultrasound-induced nucleation. Ultrasound simultaneously induces ice nucleation across the entire supercooled slurry, producing ice crystals of similar sizes. This proves a finer control of the pore size and greater reproducibility of scaffolds.

#### 4.3.5. Degree of Anisotropy against Nucleation Temperature

As shown in Figure 10, the average degree of anisotropy of the collagen scaffold decreases if the ice nucleates at a more negative nucleation temperature. This implies the formation of a more isotropic scaffold arrangement at a lower nucleation temperature. A similar trend was observed in ultrasound-induced nucleation groups. This indicates that while ultrasound can affect the nucleation and hence the pore size; there is little impact on pore anisotropy. This is consistent with Nakagawa’s findings [20].

#### 4.3.6. Percolation Diameter against Nucleation Temperature

Percolation diameter measures the size of the largest sphere penetrating the scaffold without obstruction, a valid predictor of the extent of cell transport through the scaffold. With lower nucleation temperatures, the scaffold tended to have smaller percolation diameters, as illustrated in Figure 11. The percolation diameter depends on the size, shape and number of pathways of connected pores. Scaffolds with smaller pores might tend to have narrower pathways of connected pores, impeding the movement of large objects. The application of ultrasound does not appear to impact the scaffold percolation diameter significantly. Ultrasound-assisted nucleation might allow a degree of independent control of scaffold pore size and interconnectivity.

### 4.4. Impact of Ultrasound-Trigged Nucleation on Ice Solidification and Final Scaffold Structure 

In freeze drying, the ice crystal structure is a negative template of the resulting scaffold architecture. As summarised in Figure 1, the formation of ice crystals proceeds in two stages: nucleation and growth. Nucleation is stochastic and can happen across a wide range of temperatures under the same external condition. After nucleation, the crystal growth and annealing proceed, which can be quantified by the time at equilibrium [16]. Overall, the ice crystal size is influenced by both nucleation temperature and time at equilibrium. These two thermal parameters are related: with a larger driving force (lower nucleation temperature), more nuclei are formed, limiting crystal growth (shorter time at equilibrium).

In this study, we demonstrate the ability of ultrasound to trigger ice nucleation instantaneously at certain temperatures due to the cavitation effect. As shown in Figure 13, with ultrasound, more nuclei are formed, and the crystal growth stage is shortened, forming smaller and homogeneous ice crystals. Therefore, ultrasound-induced nucleation can produce scaffolds with smaller average pore size and narrower pore size distribution. In summary, the incorporation of ultrasound into the conventional freeze-drying protocol can improve the control and consistency of scaffold architecture. We can apply ultrasound to induce nucleation at different temperatures, tuning the scaffold architecture for specific tissue engineering applications.

## 5. Conclusions

In standard freeze-drying protocols to fabricate 3D porous collagen scaffolds, ice nucleation occurs across a wide range of temperatures. The large variability in nucleation temperature of ice can lead to poor reproducibility. At a small undercooling, large, anisotropic plate-like structures are formed, while with larger undercooling, smaller and homogenous pores are formed. The stochastic nature of nucleation means that scaffolds show large structural variations even under the same fabrication protocol. Nowadays, ultrasound may be used as a method to trigger nucleation at a particular temperature. Ultrasound-induced nucleation is characterised by a sharp increase in temperature to the equilibrium temperature. Scaffolds with ultrasound-induced nucleation have a smaller average pore size and a narrower distribution of pore sizes compared with those formed with stochastic nucleation. However, the overall scaffold anisotropy and interconnectivity remained unchanged. The incorporation of ultrasound-trigged nucleation into the standard freeze-drying protocol therefore provides a route for finer control over scaffold architecture.

## Figures and Tables

**Figure 1 polymers-16-00213-f001:**
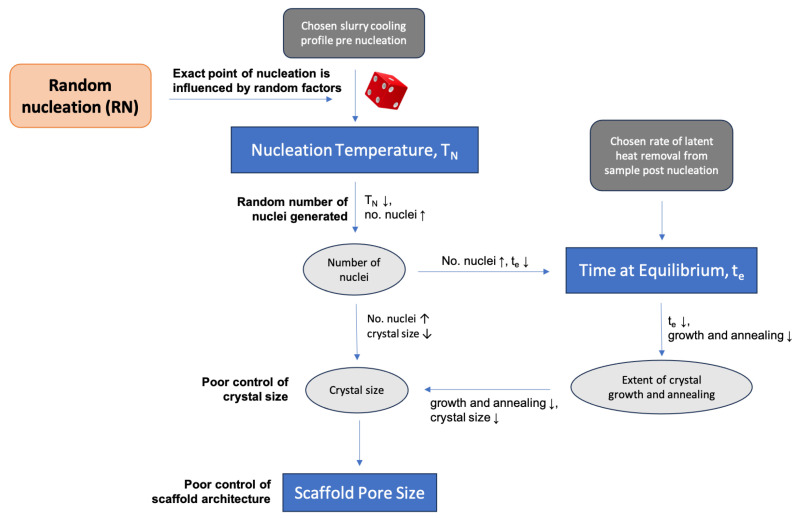
A flowchart showing how the random nucleation of ice in the freezing collagen slurry leads to poor control over scaffold architecture.

**Figure 2 polymers-16-00213-f002:**
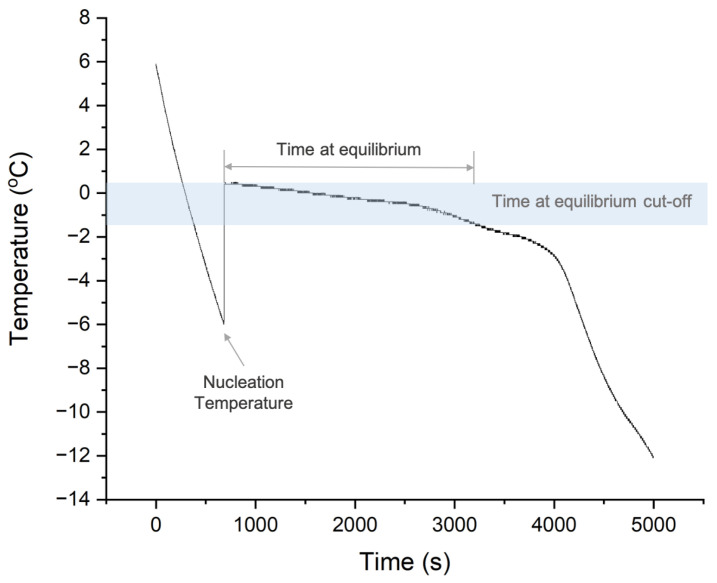
A typical thermal profile of slurry during freezing, with the nucleation temperature and time at equilibrium defined.

**Figure 3 polymers-16-00213-f003:**
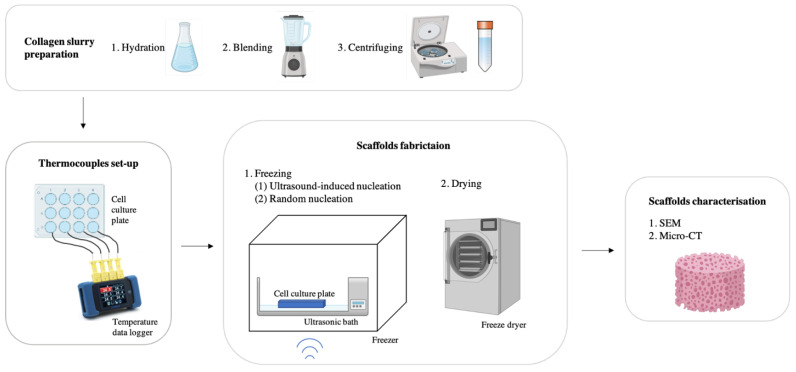
Schematic showing the experimental procedure.

**Figure 4 polymers-16-00213-f004:**
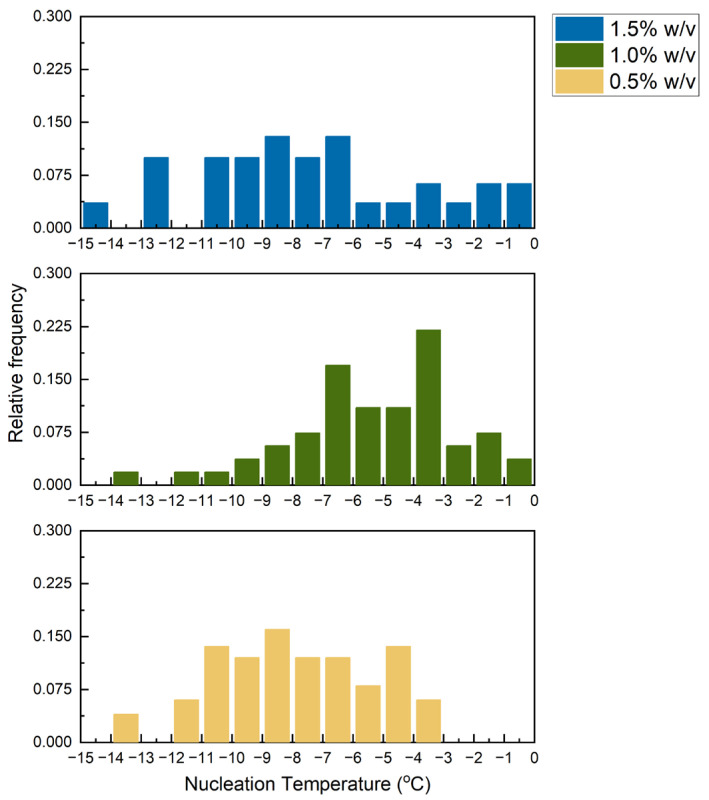
Histograms showing the distribution of nucleation temperatures of 1.5%, 1.0% and 0.5% *w/v* collagen slurry. Sample count = 39, 54 and 41, respectively. Average nucleation temperature = −7.4, −5.2 and −8.0, respectively. Range of nucleation temperature = 13.1, 13.7 and 10.1, respectively.

**Figure 5 polymers-16-00213-f005:**
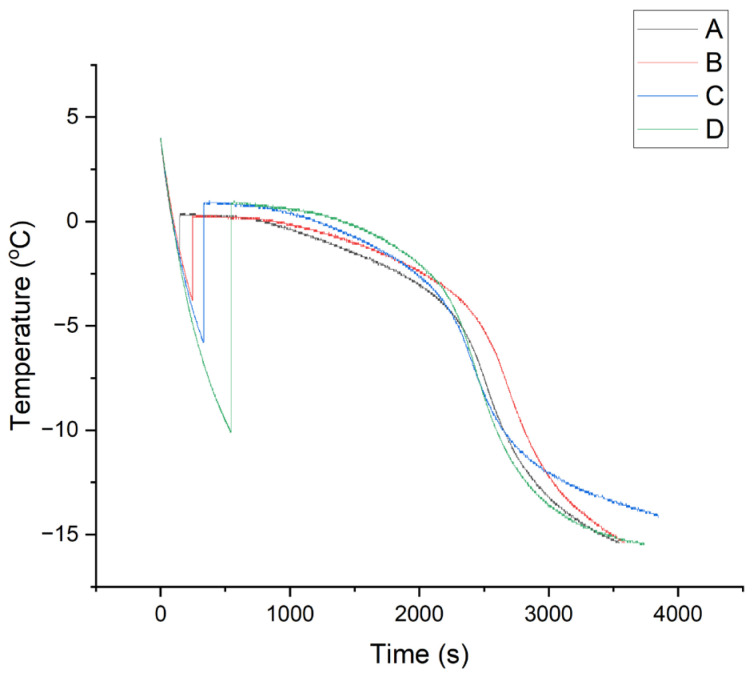
Thermal profiles of ultrasound-triggered ice nucleation and growth in 1 wt% collagen slurry. Ultrasound was applied to stimulate nucleation at (**A**) −1.7 °C, (**B**) −3.8 °C, (**C**) −5.8 °C and (**D**) −10.1 °C.

**Figure 6 polymers-16-00213-f006:**
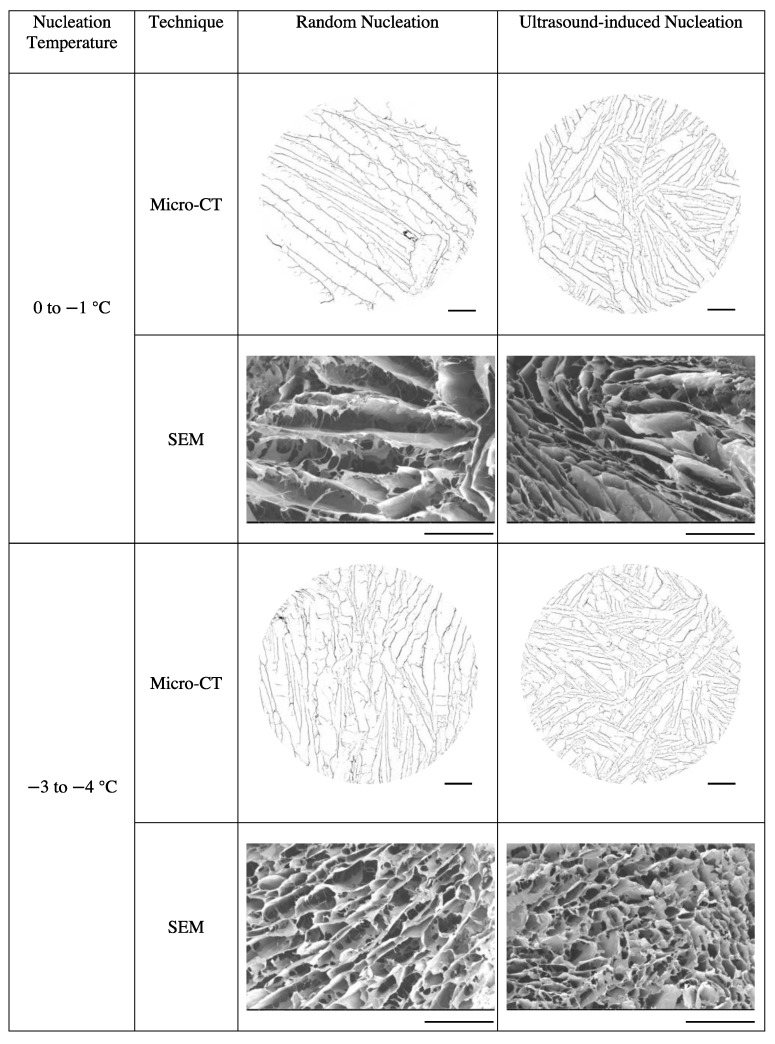
Reconstructed micro-computed tomography scan and scanning electron microscopy images of collagen scaffolds produced under random and ultrasound-induced nucleation at 0 to −1 °C, −3 to −4 °C, −6 to −7 °C and −10 to −11 °C. Images were taken at the horizontal cross-sections at the centre of the scaffold. Scale bar = 1 mm.

**Figure 7 polymers-16-00213-f007:**
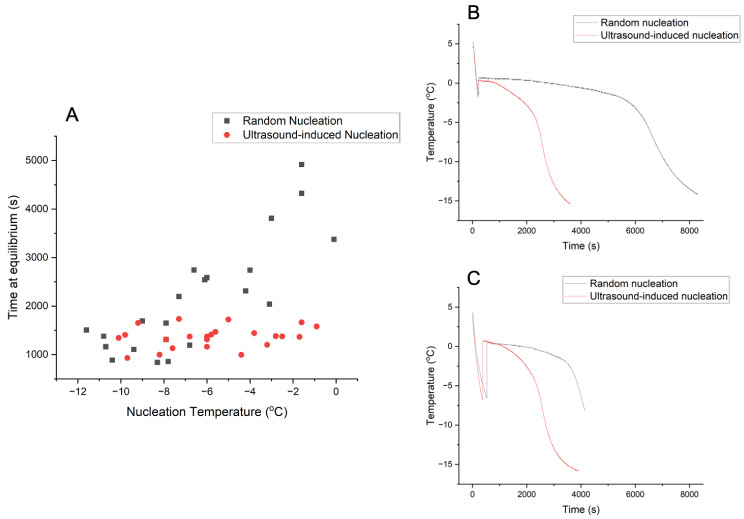
(**A**) Time at equilibrium against nucleation temperature. (**B**) Thermal profile of collagen slurries with ice nucleating at around −2 °C. (**C**) Thermal profile of collagen slurries with ice nucleating at around −6 °C.

**Figure 8 polymers-16-00213-f008:**
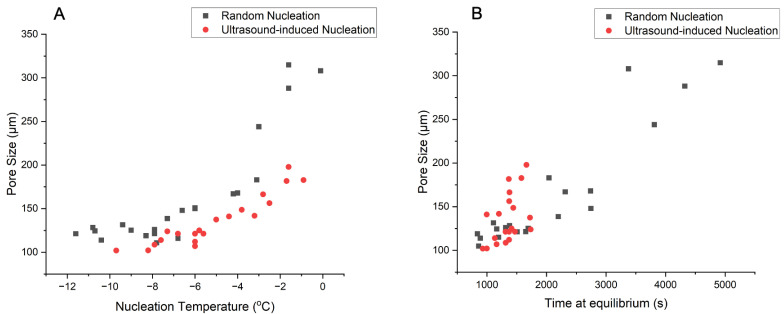
(**A**) Average scaffold pore size against nucleation temperature. (**B**) Average scaffold pore size against time at equilibrium.

**Figure 9 polymers-16-00213-f009:**
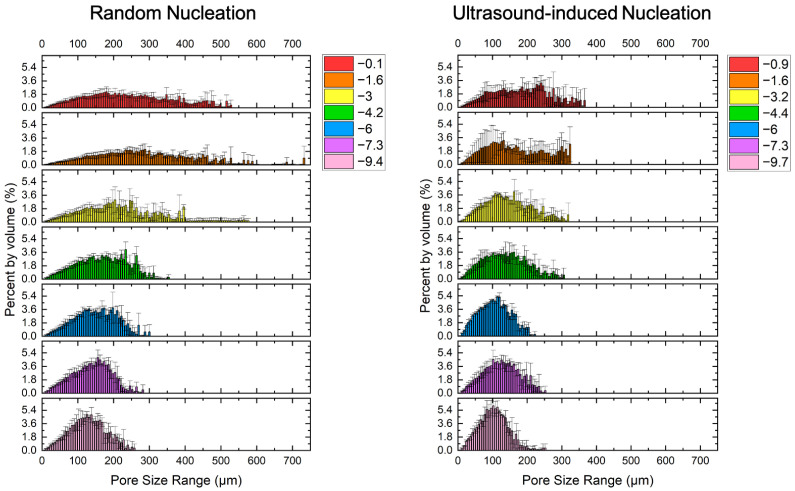
Pore size distribution of scaffolds underwent random and ultrasound-induced nucleation at different nucleation temperatures (°C). Columns and error bars represent the average percentage of pores with a pore size in any given 6 μm interval and standard deviation measured from three volumes of interest (VOIs) along the height of the scaffold.

**Figure 10 polymers-16-00213-f010:**
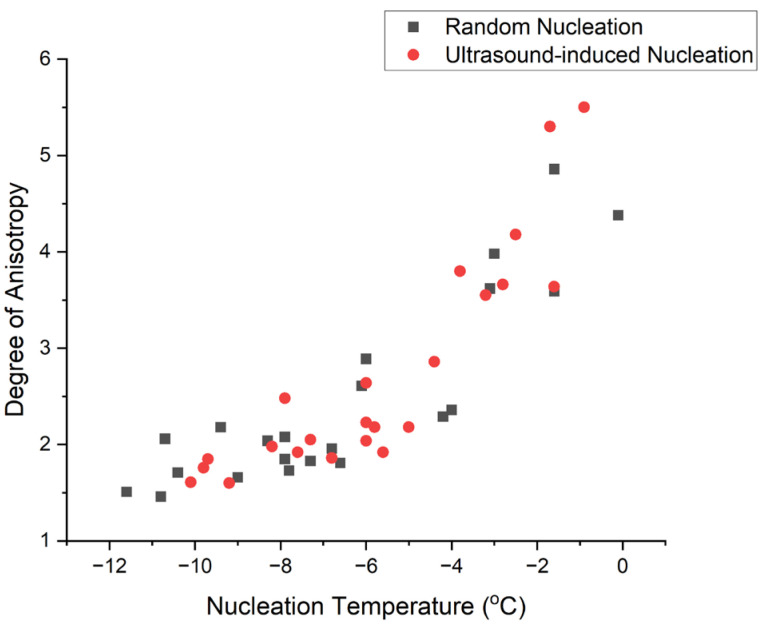
Scaffold degree of anisotropy against nucleation temperature.

**Figure 11 polymers-16-00213-f011:**
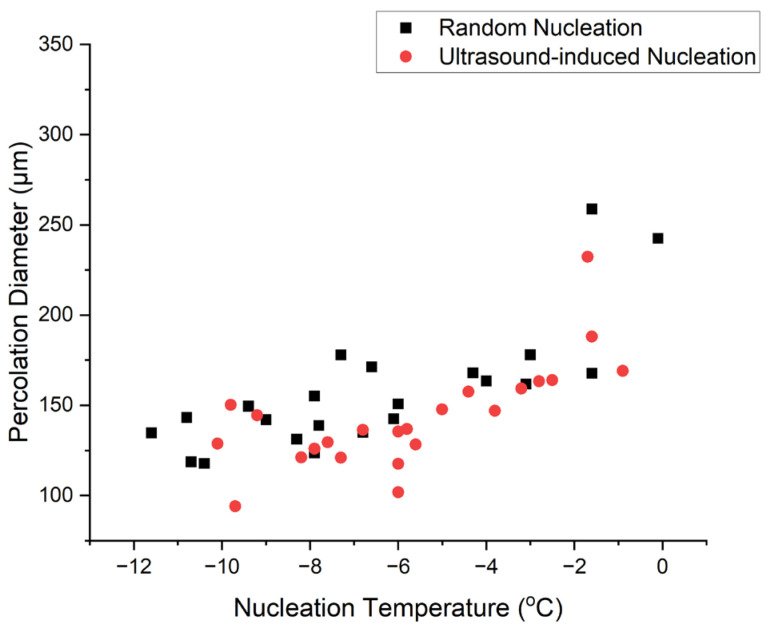
Scaffold percolation diameter against nucleation temperature.

**Figure 12 polymers-16-00213-f012:**
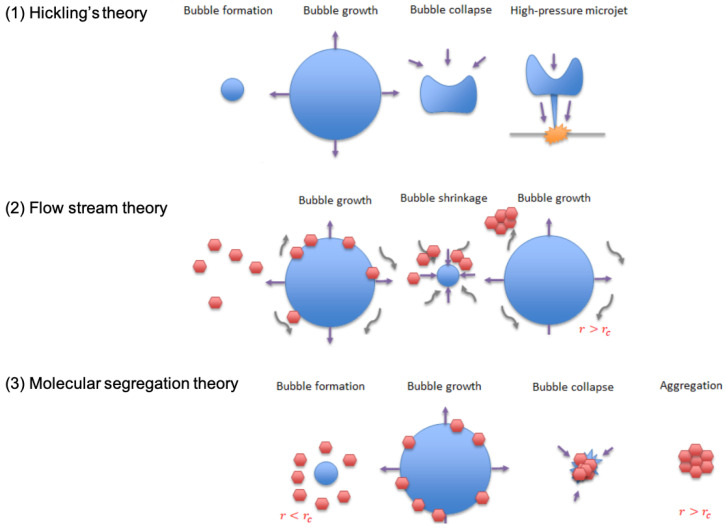
Schematic representation of gas bubble behaviour when ultrasound was applied by three different theories. (1) Hickling’s theory: graph adapted from “Nucleation of freezing by cavity collapse and its relation to cavitation damage” [27]. (2) Flow stream theory: graph adapted from “Ultrasonic-induced nucleation of ice in water containing air bubbles” [34]. (3) Molecular segregation theory: graph adapted from “The effect of ultrasound on crystallisation-precipitation processes: some examples and a new segregation model” [35]. Red hexagons represent molecules, and blue circles represent bubbles. r = radius, r_c_ = critical radius for nucleation.

**Figure 13 polymers-16-00213-f013:**
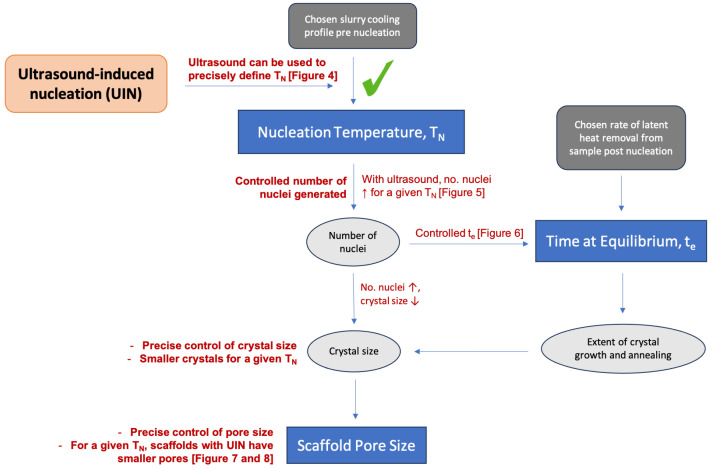
A flowchart illustrating how application of ultrasound affects the solidification process, final ice crystal size and scaffold pore structure.

## Data Availability

Our research data are openly available in Apollo at https://doi.org/10.17863/CAM.99991 (accessed on 29 November 2023), reference number 99991.

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
