# Peer review of "Controlling the Architecture of Freeze-Dried Collagen Scaffolds with Ultrasound-Induced Nucleation"

_polymers, 2024, doi:10.3390/polym16020213_

Round 1

Reviewer 1 Report

Comments and Suggestions for Authors

In this work, the low-frequency ultrasound was used to overcome the disadvantages of the stochastic process by trigger nucleation. The application of ultrasound was found to define the nucleation temperature of collagen slurries, precisely tailoring the pore architecture and providing important new structural and mechanistic insights. This paper is thorough, well-structured and likely of interest to readers focused on application of collagen in tissue engineering. The study is very interesting. In my opinion, this article can be accepted for publication in Polymers after minor revision. Questions:

1. The ‘Introduction’ part should be rearranged. It has a little chaotic.

2. Some Figures need to be revised and beautified. For example, the concentration in the box should be put in the Figure 4.  

3. The author should discuss the reaction mechanism of low-frequency ultrasound in detail in manuscript. In addition, besides collagen, does the low-frequency ultrasound can be used for the other polymer, please?

Comments on the Quality of English Language

Minor editing of English language required.

Author Response

Thanks for your time reviewing this manuscript. Based on your valuable comments and questions, we have made some modifications to the manuscript. Please find the detailed responses below and the corresponding corrections were highlighted in the revised manuscript.

Comments 1: The ‘Introduction’ part should be rearranged. It has a little chaotic.

Response 1: Thanks for your suggestion. The introduction starts with the mechanism of freeze-drying, and the principle of ice formation (nucleation and growth). Then, we tried to correlate different practical ways to tailor the scaffold architecture with the ice formation process in collagen slurry. Finally, we highlighted the poor reproducibility in scaffold fabrication and suggested ultrasound as a potential method to tackle this problem.

We have tried to rephrase some leading sentences and restructured some paragraphs (changes highlighted in yellow). Also, “Scaffold structure can be quantified by not only pore size and isotropy but also the percolation diameter. The percolation diameter is a scale-invariant property describing the transport pathway through the scaffold [18]. It is defined as the size of the largest sphere able to migrate through an infinitely large scaffold without obstructions. This parameter provides a valid prediction of the extent of cell migration through the scaffold [19], [20]” has been removed. Percolation diameter was introduced in the experimental section (lines 224-234).

Comments 2: Some Figures need to be revised and beautified. For example, the concentration in the box should be put in the Figure 4. 

Response 2: Thanks for this comment. The captions in Figure 4 have been rearranged.

Comment 3: The author should discuss the reaction mechanism of low-frequency ultrasound in detail in manuscript. In addition, besides collagen, does the low-frequency ultrasound can be used for the other polymer, please?

Response 3: Thanks for your advice. The mechanism of ultrasound-induced nucleation is explained in Section 1 (lines 94-111) and Section 4.2 (lines 353-379). More explanations are added (highlighted in yellow). The schematic of three popular mechanisms of ultrasound-induced nucleation is depicted in Figure 12. Ultrasound has been used in the fabrication of polyimide aerogels and agar gels (line 116 and 120, highlighted in blue).

Thanks again for your time and professional advice. We look forward to hearing back from you.

Sincerely,

Xinyuan Song

Department of Materials Science and Metallurgy, University of Cambridge, UK

4/Jan/2024

Reviewer 2 Report

Comments and Suggestions for Authors

The manuscript by Song and colleagues aimed to evaluate the effect of ultrasound treatment on the nucleation of type I collagen. The authors applied 40 kHz ultrasound for 3 seconds to the collagen slurry and evaluated the pore morphology, relationship between time at equilibrium with nucleation temperature, pore size against nucleation temperature and time at equilibrium, pore size at different nucleation temperatures, degree of anisotropy against nucleation temperature, and percolation diameter against nucleation temperature. Overall, the manuscript is well-written and comprehensively evaluates and explains the phenomenon. I believe that this merits the publication of this paper. With that being said, I have a few questions regarding the study, which can be found below:

1.     Regarding the ultrasound application, what is the consideration for using 40 kHz ultrasound for 3 seconds? Is this based on prior optimization or following prior studies? I believe that data showing the effect of different ultrasound frequencies (e.g., 25 kHz, 100 kHz) at same time exposure (3 s) or using 40 kHz at varying time (e.g., 3 s, 60 s, or 60 min like those reported in prior studies) can be included to elucidate the role of ultrasound frequencies and time of exposure. 

2.     There seems to be no collagen concentration-dependent trend in the average nucleation temperature as 1.0 %w/v collagen showed the highest average at -5.2 C, while 0.5 %w/v and 1.5 %w/v are relatively similar. What causes this phenomenon? Please add a few remarks regarding this point in the Discussion section.

3.     Please check and revise the typographical errors. For example: (um) in axis name of Figure 8, 9 and 11 should be changed to (µm).

Author Response

Thanks for your time reviewing this manuscript. Based on your valuable comments and questions, we have made some modifications to the manuscript. Please find the detailed responses below and the corresponding corrections were highlighted in the revised manuscript.

Comments 1: Regarding the ultrasound application, what is the consideration for using 40 kHz ultrasound for 3 seconds? Is this based on prior optimization or following prior studies? I believe that data showing the effect of different ultrasound frequencies (e.g., 25 kHz, 100 kHz) at same time exposure (3 s) or using 40 kHz at varying time (e.g., 3 s, 60 s, or 60 min like those reported in prior studies) can be included to elucidate the role of ultrasound frequencies and time of exposure.

Response 1: Many thanks for this question. Common frequencies for ultrasonic cleaning are 24 kHz and 40 kHz, as this lower range of frequency allows for more sustained pressure due to greater wavelengths, with increased penetration. Conversely, higher frequencies (exceeding 2 MHz) are regularly used in diagnostic ultrasound imaging due to their lower wavelengths, which allow for high image resolution at lower penetration depths. So, we selected 40 kHz for the purpose of this study. 3s exposure time is chosen because this period is sufficient to induce stable nucleation, and not inducing noticeable heat to the collagen slurry (temperature monitored by the thermocouple). Long ultrasound exposure time can build up heat, possibly causing negative thermal effects to the collagen slurry (denaturation).

It will be really interesting to learn the effect of different ultrasound frequencies on ice nucleation in collagen slurry. However, our ultrasonic bath has a pre-set frequency of 40 kHz. Introducing another ultrasonic bath with different frequencies may introduce some uncontrolled variables to the experiments (e.g., different locations of ultrasound power source and different thermal environments inside the bath).

Comments 2: There seems to be no collagen concentration-dependent trend in the average nucleation temperature as 1.0 %w/v collagen showed the highest average at -5.2 C, while 0.5 %w/v and 1.5 %w/v are relatively similar. What causes this phenomenon? Please add a few remarks regarding this point in the Discussion section.

Response 2: Thanks for this advice. Explanation has been incorporated in the discussion section (lines 343-350, highlighted in yellow). “The distribution of nucleation temperatures in the 1 wt% collagen slurry was centred around a higher temperature than samples based on a 0.5 wt% slurry. The higher concentration of solids provides more nucleation sites, reducing the amount of undercooling required for nucleation. However, in 1.5 wt% collagen slurries, the increased viscosity is likely to have suppressed the formation of nuclei [46]. This effect might outweigh the effects of the presence of increased nucleation sites, shifting the nucleation events to higher degrees of undercooling.”

Comment 3: Please check and revise the typographical errors. For example: (um) in axis name of Figure 8, 9 and 11 should be changed to (µm).

Response 3: We are sorry for the typographical errors. The name of y-axis in Figure 8, 9 and 11has been corrected.

Thanks again for your time and professional advice. We look forward to hearing back from you.

Sincerely,

Xinyuan Song

Department of Materials Science and Metallurgy, University of Cambridge, UK

4/Jan/2024
